# Stoic attitude in patients with cancer from the NEOcoping study: Cross-sectional study

**David Gomez[1], Alberto Carmona-Bayonas[2], Raquel Hernandez[3], Oliver Higuera[4], Jacobo Rogado[5], Vilma Pacheco-Barcia[6], María Valero[7], Mireia Gil-Raga[8], Mª Mar Muñoz[9], Rafael Carrión-Galindo[10], Paula Jimenez-Fonseca[11], Caterina Calderon[12]***

1 Department of Medical Oncology, Hospital Universitario Central of Asturias, IPSA, Oviedo, Spain,
2 Department of Hematology and Medical Oncology, Hospital Universitario Morales Meseguer, Instituto Murciano de Investigación Biosanitaria (IMIB), UMU, Murcia, Spain, 3 Department of Medical Oncology, Hospital Universitario de Canarias, Tenerife, Spain, 4 Department of Medical Oncology, Hospital Universitario La Paz, Madrid, Spain, 5 Department of Medical Oncology, Hospital Universitario Infanta Leonor, Madrid, Spain, 6 Department of Medical Oncology, Hospital Central de la Defensa Gomez Ulla, Madrid, Spain, 7 Department of Medical Oncology, Hospital Quirón salud Sagrado Corazón, Sevilla, Spain, 8 Department of Medical Oncology, Consorcio Hospital General Universitario de Valencia, Valencia, Spain, 9 Department of Medical Oncology, Hospital Virgen de La Luz, Cuenca, Spain, 10 Department of Medical Oncology, Hospital Universitario del Sureste, Arganda del Rey-Madrid, Spain, 11 Universidad del País Vasco (UPV/EHU), País Vasco, Spain, 12 Department of Clinical Psychology and Psychobiology, Faculty of Psychology, University of Barcelona, Barcelona, Spain

* ccalderon@ub.dedu

**Data Availability Statement:** All relevant data for this study are within the paper and its Supporting information files. Additional data are available from

## Abstract

### Aim

Stoicism has been applied to describe a wide range of behaviors in the face of disease and influences an individual's use of coping strategies. This study tested the relationship between stoicism and social support, optimism, psychological distress, and coping strategies in patients with cancer.

### Method

NEOcoping is a multicenter, cross-sectional study. Participants' data were collected using a standardized, self-report form and LSS, MSPSS, Mini-MAC, BSI-18, and LOT-R questionnaires. Linear regression analyses were used to assess the association between stoicism and distress scores in both genders. A total of 932 individuals with non-metastatic, resected cancer were recruited.

### Results

Males perceived a higher risk of recurrence and toxicity with adjuvant chemotherapy and obtained higher stoic attitude scores than females. Women scored higher on somatization, depression, and anxiety. Patients with high stoicism scores were older and experienced more maladaptive coping (helplessness, anxious preoccupation), and depression, while those with lower stoicism scores had greater perceived social support, optimism, and positive attitude. In both males and females, stoicism correlated negatively with perceived social support, optimism, and positive attitude, and positively with helplessness, anxious

www.neocoping.es for researchers who meet the criteria for access to confidential data.

**Funding:** This work was supported by the FSEOM-Onvida 2015 grant for Projects on Long Survivors and Quality of Life. The sponsor of this research has not participated in data collection, analysis, or interpretation, in writing the report, or in the decision to submit the article for publication.

**Competing interests:** he authors have declared that no competing interests exist.

preoccupation, and depression. In men, stoicism was directly and negatively associated with social support and optimism, and positively with anxious preoccupation. In women, stoicism was positively associated. In women, stoicism was directly and negatively associated with social support and positively with age and optimism. Stoicism was directly and positively associated with helplessness.

## Discussion

A stoic attitude was associated with lower social support, reduced optimism, and passive coping strategies (helplessness and anxious preoccupation) in this series of patients with cancer.

## Introduction

Stoicism, the school of philosophy created by Zeno of Citium (332–263 BC), is one of the philosophical doctrines with the most rebirths throughout history, influencing numerous modern, psychological, humanistic, and moral currents of thought [1]. Originally, stoicism sought a balanced solution to the dichotomy of determinism-free will, advocating a virtue that resided in knowledge of the laws of the cosmos and pragmatic acceptance of them. However, in its modern psychological conception, stoicism has come to denote a construct based on the suppression of all emotional expression, increased tolerance for suffering, and the acceptance of the vicissitudes of life without complaint [2].

Despite the potential explanatory capacity of different attitudes, stoicism has been ignored in health-related spheres [1]. One of the causes lies in the cultural and personality-based parameters that define stoicism. Spiers' theory of stoicism has provided a model for describing a wide range of behaviors in the face of disease ranging from silence, resilience to adversity, or being able to endure pain without complaint [3, 4]. For Spiers, stoic traits would be adaptive in the area of health, as they would increase self-efficacy by perceiving one's own limits and prompting the adoption of proactive measures to control symptoms through the use of reason [3, 4]. Some studies have shown that stoicism would be very sensitive to the health setting, and this could have implications for the patient's treatment and prognosis [5, 6]. Maintaining a stoic attitude towards pain and knowing one's own limits could be particularly beneficial for the patient and allow them to adopt a more adaptive coping style [3], or conversely, the stoic attitude could restrict attempts at medical intervention if the patient does not report their real clinical situation (i. e., the patient feels pain, but does not show it). Therefore, according to other researchers, stoic characteristics hinder seeking care, delay diagnosis, and prevent the implementation of improvements in psychological care [1].

In oncology, an issue worth exploring is whether a stoic attitude has an influence, not only in terminal disease [7–9], but also in the early stages of localized, potentially curable cancers. As cancer is a stressful event associated with fear and prognostic uncertainty, it is common for patients to employ coping strategies to deal with it. Stoicism and emotional distress (anxiety, depression, and somatization) may interfere with coping by increasing anxious preoccupation [10]. Dispositional optimism may act as a resource that maintains a positive mood, provides a more flexible coping style to face with the inability to control stressful stimuli, and protects patients from the possible negative effects of cancer treatment and cancer itself in case of recurrence [11, 12]. Likewise, social support, defined as the perception that others are willing to help, appears to operate similarly to optimism and is a critical factor in decreasing negative

psychological affect in individuals with cancer, which can contribute to better adjustment to the disease [13].

Overall, the interrelationships between social support, optimism, stoicism, coping strategies and psychological distress remain to be further investigated. In the oncology field, little has been explored about the pathways through which stoicism influences coping in cancer patients. Some authors suggest that men and older people have higher levels of stoicism than women [2, 14] or younger people [1]. Other authors have suggested that men and older adults are more stoic because they find it more difficult to identify and express their emotions [15], which would have a negative influence on seeking psychological help [16]. Meanwhile, other research suggests that the stoic attitude is more related to the ability to self-regulate one's emotions [17].

In this perspective, our study has assessed the fit of an adapted stoicism model [3, 4] using data collected from a large sample of patients who had cancer resected with curative intent. This adapted model of stoicism represents key constructs (social support, optimism, psychological distress, stoicism, and coping strategies), as well as the relationships between the variables used to operationalize the constructs. We presume that: a) social support and optimism will positively affect stoicism and coping strategies; b) stoicism will be related to coping strategies; and c) the relationships between social support, optimism and coping strategies will be mediated by stoicism.

## Methods

### Study design and population

NEOcoping is a cross-sectional, observational, and multicenter study sponsored by the Continuous Care Group of the Spanish Society of Medical Oncology (SEOM). Patients were recruited consecutively in eleven Spanish university hospitals between 2017 and 2019. After providing written informed consent, eligible patients were included during their first visit to a medical oncology department, which they attended in the month following resection of non-metastatic cancer. The structure of the first visit followed the standard practice of each center, requiring, at the very least, that the diagnosis, prognosis or risk of cancer recurrence, and therapeutic alternatives, with their incumbent risks of toxicity and benefits be explained. Once the decision whether or not to receive chemotherapy was made, the subject was invited to participate in this study, making the decision to accept systemic adjuvant treatment completely independent of recruitment. The ethics review board at each of the participating institutions and the Spanish Agency for Medicines and Medical Devices (AEMPS) approved the study and all procedures were in accordance with the Helsinki Declaration.

Inclusion criteria were ≥18 years of age with resected, non-metastatic cancer for which the clinical practice guidelines recommended adjuvant chemotherapy. Exclusion criteria were cognitive impairment, any neoadjuvant therapy, adjuvant treatment consisting solely of radiotherapy, hormone therapy or other treatment modalities other than chemotherapy, any contraindication for adjuvant chemotherapy (poor functional status, age, comorbidities, etc.). Of the 1.016 patients screened, 84 were not eligible (23 did not meet inclusion criteria, 28 met exclusion criteria, and data were missing for 33). A total of 932 individuals with non-metastatic, curable resected cancer were recruited.

### Instruments and data collection

Data were collected from the individuals and from clinical records. Psychometric questionnaires were completed by the patients at home and delivered at the next medical visit during which the oncologist clarified any doubts surrounding the questions in the questionnaires.

Participants were informed that the answers would be treated anonymously and that there were no "right" or "wrong" answers, with the aim of minimizing the risk of random response. Data were collected through the centralized web platform www.neocoping.es that controls for lost data, inconsistencies, and errors in real time, and double-checks the study variables, with telephone and on-line monitoring (PJF).

**Liverpool Stoicism Scale (LSS)** is a self-report questionnaire designed to measure stoicism. It contains 20 items, including lack of emotional involvement, dislike for openly expressing emotion, and the ability to withstand emotion. The Spanish version was translated and validated by our group [6]. Responses were recorded on a 5-point Likert scale ranging from *strongly disagree* to *strongly agree*, yielding a sum score of between 20 and 100. Higher scores indicate greater stoicism. The scale has good internal consistency (Cronbach's alpha = 0.83) [6].

**Multidimensional scale of perceived social support (MSPSS)** is a brief scale that assesses three sources of support: family, friends, and significant other. Items were measured on a 5-point Likert scale [18]. A higher score indicates more perceived support. The scale in this study has good internal consistency (Cronbach's alpha = 0.90).

**Mini-Mental Adjustment to Cancer (Mini-MAC)** scale is one of the most widely used instruments to quantify coping responses in individuals with cancer [19]. The 29-item mini-MAC assesses five cognitive coping responses: helplessness (e.g., '*Can't handle it*'), anxious preoccupation (e.g., '*I am apprehensive*'), fighting spirit, fatalism (e.g., '*At the moment I take one day at a time*'), and cognitive avoidance (e.g., '*Not thinking about it helps me cope*'). For this study, the version with a 4-factor structure validated in Spanish was used, grouping fighting spirit and fatalism into a single factor, positive attitude [20]. Each item is scored from 1 to 4. High scores indicate that coping styles are used more often. Satisfactory reliability was found for scales, Cronbach's alpha ranged from 0.52 to 0.90 [21].

**Brief Symptom Inventory (BSI-18)** includes 18 symptoms to evaluate the degree of distress on a 5-point Likert scale [22]. The scale groups symptoms into somatization, depression, and anxiety. Cronbach's alpha ranged from 0.81 to 0.90 [23].

**Revised Life Orientation Test (LOT-R)** consists of optimism and pessimism scales, with ten items (three items assess optimism, three evaluate pessimism, and four are fillers). Respondents indicated the extent to which they agreed with each item on a 5-point Likert scale. In this study, Cronbach's alpha ranged between 0.74 and 0.78.

Patients were asked to predict their risk of cancer recurrence with and without chemotherapy, as well as their risk of severe toxicity with chemotherapy, using a numerical rating scale from 0 to 100.

**Demographic data.** The following data were obtained with respect to the sample's medical and demographic characteristics. Patient characteristics included gender; age; marital status; educational level; occupational field; tumor site, stage, and time since diagnosis. Clinical estimations were obtained from the medical oncologist who cared for the patient at their first appointment to assess the suitability of administering adjuvant treatment after surgery with curative intent.

## Data analysis

Demographic information was reported by frequency analyses. Gender and stoicism differences on age, perception of life expectancy, and risk of toxicity, and psychological scales were measured using independent samples t-tests. Chi-squared analyses were used to investigate potential differences between men and women. Effect sizes were calculated by Cohen's statistics. Cohen's d was reported as an indicator of the effect size of the differences, with d > 0.2

representing a small, d > 0.5 a medium and d > 0.8 a large effect size [24]. Bivariate correlations were calculated between stoicism (LSS) and psychological scales (MSPSS, Mini-Mac, BSI-18, and LOT-R) and sociodemographic factors. All data were checked for normality, outliers, and the assumptions of multicollinearity and homoscedasticity. Structural Equation Modeling (SEM) is a method for building, estimating, and testing theoretical models of the relationships between variables. It can be used instead of multiple regression and other methods to analyze the strength of correlations between individual variable indicators in a specific population [25]. In this study, the previously determined significant factors were used in the SEM to identify the relationship between stoicism and psychological scales. Standardized direct, indirect, and total effects with corresponding 95% bias-corrected confidence intervals (CI) were measure using the bootstrapping methods [25, 26]. The model fit was tested using the normed $\chi^2$ value (NC; desired value < 2.0, desired significance $P > 0.05$), the goodness-of-fit index; the Comparative Fit Index (CFI), the Tucker-Lewis index (TLI), the Normed Fit index (NFI) (>0.95 indicating an excellent fit); root mean square of approximation (RMSEA; desired value< 0.06) [25]. For all the tests carried out, bilateral statistical significance was set at $p < .05$. Statistical analyses were performed using the IBM-SPSS 23.0 statistical and AMOS 23.0 software package for Windows PC.

## Results

### Patients' baseline characteristics

Mean age was 62.1 years (standard deviation [SD] = 12.1, range 24–85), 61% (n = 566) were female and 39% (n = 366) were male. Baseline characteristics are presented in Table 1. The most frequent tumor types were adenocarcinomas of the colon (42%), breast (34.4%), and stomach (4.5%). Approximately half (43.8%) were stage III. Most subjects were married or living with a partner (76.3%), had at least primary education (53.6%), and claimed to be unemployed or retired (58.0%). The estimated risk of relapse with chemotherapy was 21.1%, and 37.7% without chemotherapy.

### Gender differences with respect to age, perception of risk of recurrence and toxicity, stoicism, and psychological scales in patients with cancer

Overall, 27% of patients had low (PD≤ 46), and 33% had high stoicism scores (PD≥64). The men in this sample were older (t = 6.549; p<0.001, Cohen's $d$ = 0.43) and estimated a higher risk of recurrence without chemotherapy (t = 2.711; p = 0.007, Cohen's $d$ = 0.19) and lower risk of toxicity with chemotherapy (t = -3.717; p = 0.001, Cohen's $d$ = 0.25) than women. No differences were found between the two genders in the perception of the risk of recurrence with chemotherapy, Table 2.

Males scored higher for stoic attitude than women (t = 8.388; p<0.001, Cohen's $d$ = 0.56). Women, on the other hand, scored higher for somatization, depression, and anxiety (t = -4.547, t = -5.802 and t = -3.002; all p<0.001), as well as for anxious preoccupation (t = -3.554; p<0.001, Cohen's $d$ = 0.24). The remaining differences were not statistically significant (p>0.05). In terms of primary cancer, patients with colon cancer scored higher on stoicism than patients with breast cancer (M: 57.4 vs M 53.8; F = 10.445, p = 0.001; IJ: 3.544, p = 0.001, respectively).

### Stoicism differences by age, perception of risk of recurrence and toxicity, and psychological scales in patients with cancer

Of the 932 subjects, 33.5% (n = 312) scored high in stoicism (percentile>75), and 26.8% (n = 250) low (percentile<25) on the LSS. Patients with high stoicism scores were older than

**Table 1. Patient characteristics (n = 932).**

| Characteristic | N | % |
|---|---|---|
| Gender | | |
| *Female* | 566 | 60.7 |
| *Males* | 366 | 39.3 |
| Age, years | | |
| *Less than 45* | 144 | 15.5 |
| *46–55* | 203 | 21.8 |
| *56–65* | 265 | 28.4 |
| *More than 66* | 320 | 34.3 |
| Cancer type | | |
| *Colon* | 391 | 42.0 |
| *Breast* | 321 | 34.4 |
| *Stomach* | 42 | 4.5 |
| *Others* | 178 | 19.1 |
| Cancer stage | | |
| *I* | 130 | 14.0 |
| *II* | 390 | 41.9 |
| *III* | 408 | 43.8 |
| *Missing* | 3 | 0.3 |
| Educational level | | |
| *Less than high school* | 500 | 53.6 |
| *More than high school* | 432 | 46.4 |
| Marital status | | |
| *Married/ partnered* | 710 | 76.3 |
| *Single* | 80 | 8.4 |
| *Separated/ divorced* | 79 | 8.5 |
| *Widowed* | 63 | 6.8 |
| Employment status | | |
| *Full or part-time* | 540 | 58.0 |
| *Retired, homemaker, unemployed* | 392 | 42.0 |

those who scored lower on this domain; mean age 64.4 versus 61.4 years, respectively (t = -6.941; p<0.001, Cohen's *d* = 0.25), and more depressed (t = -2.609; p = 0.009, Cohen's *d* = 0.23) Table 3. As for individuals' estimation of the risk of relapse and toxicity with chemotherapy based on their level of stoicism, no statistically significant differences were found. Participants with lower stoicism scores had greater perceived social support, positive attitude, and higher levels of optimism than more stoic individuals. On the other hand, subjects with high stoicism scores had more maladaptive coping based on helplessness and anxious preoccupation than patients with low scores. The remaining differences between the two groups were not statistically significant, Table 3.

## Relation between stoicism and social and psychological scales in men and women

Correlations were calculated between stoicism and the main study variables in men and women (Table 4). Bivariate correlations between LSS and psychological scales were inconsistent with the notion of stoicism as a factor in psychological resilience. In males, stoicism correlated negatively with perceived social support, coping strategy (positive attitude), and

**Table 2. Gender differences in age, perception of risk of recurrence and toxicity, stoicism, and psychological scales in patients with cancer.**

| Variables | Men (n = 366) | | Women (n = 566) | | t-value | |
| --- | --- | --- | --- | --- | --- | --- |
| | Mean | SD | Mean | SD | T | P |
| Age (years) | 62.1 | 12.1 | 56.9 | 11.8 | 6.549 | 0.001* |
| Perception of risk of: | | | | | | |
| _Recurrence with CT_ | 78.6 | 22.1 | 78.1 | 21.6 | 0.320 | - - - |
| _RRecurrence without CT_ | 36.7 | 21.4 | 32.4 | 22.5 | 2.711 | 0.007* |
| _CT-related toxicity_ | 53.7 | 22.7 | 59.4 | 22.4 | -3.717 | 0.001* |
| LSS. Stoicism | 58.4 | 7.2 | 54.2 | 7.7 | 8.388 | 0.001* |
| MSPSS. Social support | 74.7 | 9.3 | 75.7 | 8.9 | -1.634 | - - - |
| Mini-MAC. Helplessness | 19.8 | 19.1 | 19.2 | 19.5 | 0.536 | - - - |
| Mini-MAC. Anxious preoccupation | 36.7 | 21.6 | 42.2 | 23.6 | -3.554 | 0.001* |
| Mini-MAC. Positive attitude | 74.5 | 17.1 | 76.1 | 15.7 | -1.404 | - - - |
| Mini-MAC. Cognitive avoidance | 51.5 | 26.3 | 54.8 | 26.0 | -1.887 | - - - |
| BSI-18. Somatization | 59.7 | 6.7 | 61.8 | 6.9 | -4.547 | 0.001* |
| BSI-18. Depression | 59.2 | 6.7 | 61.5 | 6.0 | -5.802 | 0.001* |
| BSI-18. Anxiety | 59.7 | 7.0 | 63.0 | 6.9 | -7.384 | 0.001* |
| LOT-R. Optimism | 20.7 | 3.7 | 21.1 | 4.2 | -1.478 | - - - |

_Abbreviations_: SD, Standard Deviation; CT, chemotherapy; LSS, Liverpool Stoicism Scale; MSPSS, Multidimensional Scale of Perceived Social Support; Mini-MAC, Mini-Mental Adjustment to Cancer; BSI, Brief Symptom Inventory; LOT-R, Life Orientation Test-Revised

* Statistically significant differences.

**Table 3. Comparison of cancer patients with high and low stoicism by age, perception of risk of recurrence and toxicity, and psychological scales.**

| Variables | Low stoicism (PC <25) | | High stoicism (PC >75) | | t-test for equality of means[a] | |
| --- | --- | --- | --- | --- | --- | --- |
| | n = 250 | | n = 312 | | | |
| | Mean | SD | Mean | SD | t | P |
| Age (years) | 64.4 | 12.1 | 61.4 | 11.6 | -6.941 | 0.001* |
| Perception of risk of: | | | | | | |
| _Recurrence with CT_ | 78.2 | 21.4 | 78.9 | 22.1 | -0.412 | - - - |
| _Recurrence without CT_ | 32.9 | 23.3 | 36.8 | 23.6 | -1.932 | - - - |
| _Toxicity with CT_ | 58.7 | 23.1 | 57.4 | 23.0 | 0.673 | - - - |
| MSPSS. Social support | 78.7 | 6.4 | 71.9 | 10.6 | 8.857 | 0.001* |
| Mini-MAC. Helplessness | 13.8 | 16.9 | 20.8 | 19.9 | -4.393 | 0.001* |
| Mini-MAC. Anxious preoccupation | 39.1 | 22.6 | 42.4 | 24.5 | -2.610 | 0.009* |
| Mini-MAC. Positive attitude | 78.6 | 14.8 | 70.9 | 17.8 | 5.498 | 0.001* |
| Mini-MAC. Cognitive avoidance | 51.6 | 26.8 | 52.3 | 26.1 | -0.325 | - - - |
| BSI-18. Somatization | 61.6 | 6.9 | 61.3 | 7.2 | 0.487 | - - - |
| BSI-18. Depression | 60.2 | 5.5 | 61.6 | 6.5 | -2.609 | 0.009* |
| BSI-18. Anxiety | 62.1 | 7.7 | 62.3 | 8.3 | -0.229 | - - - |
| LOT-R. Optimism | 22.1 | 4.2 | 20.0 | 3.9 | 5.990 | 0.001* |

_Abbreviation_: PC, percentile, SD, Standard Deviation; CT, chemotherapy; LSS, Liverpool Stoicism Scale; MSPSS, Multidimensional Scale of Perceived Social Support; Mini-MAC, Mini-Mental Adjustment to Cancer; BSI, Brief Symptom Inventory; LOT-R, Life Orientation Test-Revised.

[a] Before t-test analysis, Levene's test for equality of variances was applied.

* Statistically significant differences.

**Table 4. Correlation between stoicism, perception of risk of recurrence and toxicity, psychological scales in men and women with cancer.**

| Variables** | Men | | Women | |
|---|---|---|---|---|
| | r | p | r | p |
| Age (years) | 0.044 | 0.398 | 0.261 | 0.001* |
| Perception of risk of: | | | | |
| Recurrence with CT | -0.031 | — | 0.020 | — |
| Recurrence without CT | 0.139 | 0.008* | -0.050 | — |
| Toxicity with CT | -0.051 | — | -0.030 | — |
| MSPSS. Social support | -0.304 | 0.001* | -0.286 | 0.001* |
| Mini-MAC. Helplessness | 0.130 | 0.013* | 0.189 | 0.001* |
| Mini-MAC. Anxious preoccupation | 0.199 | 0.001* | 0.101 | 0.018* |
| Mini-MAC. Positive attitude | -0.198 | 0.001* | -0.149 | 0.038* |
| Mini-MAC. Cognitive avoidance | 0.018 | — | 0.079 | — |
| BSI-18. Somatization | 0.041 | — | 0.012 | — |
| BSI-18. Depression | 0.176 | 0.001* | 0.144 | 0.001* |
| BSI-18. Anxiety | 0.133 | 0.011* | 0.052 | — |
| LOT-R. Optimism | -0.276 | 0.001* | -0.257 | 0.001* |

*Abbreviations*: SD, Standard Deviation; LSS, Liverpool Stoicism Scale; MSPSS, Multidimensional Scale of Perceived Social Support; Mini-MAC, Mini-Mental Adjustment to Cancer; BSI, Brief Symptom Inventory; LOT-R, Life Orientation Test-Revised.

* Statistically significant differences

**Criterion variable: stoicism according to LSS scale.

optimism, and positively with maladaptive coping (helplessness and anxious preoccupation), anxiety, depression and estimation of the risk of recurrence without chemotherapy. In women, stoicism correlated negatively with perceived social support, positive attitude and optimism, and positively with age, coping strategies (helplessness, and anxious preoccupation), and depression.

## Trajectory analysis results: Relationship among stoicism and psychological variables

In men, as displayed in Fig 1, social support and optimism were directly, and negatively associated with stoicism ($\beta$ = -0.17 and $\beta$ = -0.23, respectively). Stoicism was directly, and positively

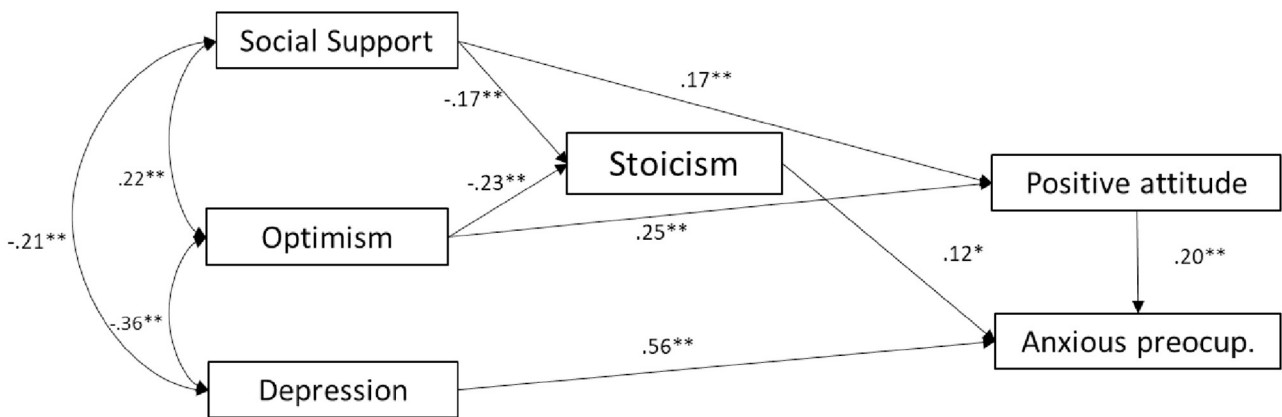

**Fig 1. Tested study model: Relationship between stoicism and psychological variables in men.** Note: Dashed lines indicate statistically non-significant paths. *p<0.01; **p<0.001.

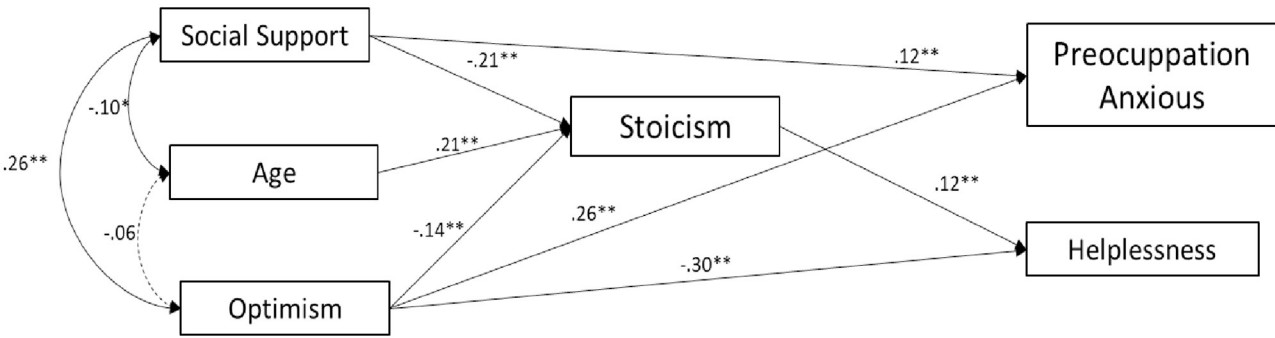

**Fig 2. Tested study model: Relationship between stoicism and psychological variables in women.** Note: Dashed lines indicate statistically non-significant paths. *p<0.01; **p<0.001.

associated with anxious preoccupation ($\beta$ = 0.12). Social support was positively associated with positive aptitude ($\beta$ = 0.17), and depression was positively associated with anxious preoccupation ($\beta$ = 0.56). The model fitted the data perfectly, $X^2$ = 11.318, p = 0.045; CFI = 0.97; TLI = 0.95; NFI = 0.96; RMSEA = 0.05 (90% CI [0.001, 0.074]).

In women, as displayed in Fig 2, stoicism was directly, and negatively associated with social support and positively with age ($\beta$ = -0.21 and $\beta$ = 0.21, respectively). Optimism was directly, and negatively associated with stoicism ($\beta$ = -0.14), and positively with anxious preoccupation and helplessness ($\beta$ = 0.26 and $\beta$ = -0.30, respectively). Stoicism was directly, and positively associated with helplessness ($\beta$ = 0.12). The model fitted the data perfectly, $X^2$ = 14.447, p = 0.044; CFI = 0.98; TLI = 0.95; NFI = 0.97M RMSEA = 0.04 (90% CI [0.000, 0.064]).

## Discussion

This study examined the relationships among stoicism, social support, optimism, psychological distress and coping in patients with a resected cancer, as proposed in the adapted stoicism model. Our results validated the findings of previous studies showing that stoicism was more a risk factor than a protective one [6, 27], since more stoic individuals perceived less social support, less optimism, and used passive coping strategies as anxious preoccupation and helplessness. This led the high stoicism-scoring participants to adapt worse to their disease. These results endorse the notion of Pinnock et al., who pointed out that stoicism could potentially be a maladaptive behavior with negative attitudes toward seeking psychological help [28].

In line with earlier studies, stoicism and psychological suffering were more common in males than women [1, 2, 28–30]. In our series, the males were older, with little age dispersion [mean age, 62.1; mode, 68]; colon was the most frequent cancer. In contrast, among the women, there were two age groups, a younger one in which breast cancer was the most common tumor site and another, older group [mean age, 56.9; mode, 48] with predominance of breast and colon cancer. As such, stoicism was unrelated to age in the men insofar as most of them were older, whereas age did impact stoicism in women. Younger females were less stoic than older ones, as reflected in the literature [1]. It is possible that stoicism increases with development in adults and age-related maturity, given the acceptance that loss is inevitable. Likewise, some authors have explained it as a cohort effect; i.e., that older people have grown up in a culture in which giving up, resignation, and stoic attitudes are particularly valued [1]. Similarly, some authors suggest that males are more stoic because it is more difficult for them

to identify and express their emotions or that they are afraid of appearing weak and 'unmanly' in light of their illness [1, 16, 31]. These differences in the characteristics of the sample are also reflected in stoicism, with colon cancer patients having higher scores on stoicism than breast cancer patients who were mostly female and younger.

Despite these discrepancies, there are more similarities than differences between men and women who have high stoicism scores. The most salient of them being that they express having less social support and are more pessimistic. Thus, it may be that if the stoic does not articulate much what is happening to them, the people around them adapt to this style and ask fewer questions to 'not to bother them' and the patient perceives it as a lack of support.

We did not detect any relationship between stoicism and the perception of risk of relapse or of toxicity. Nevertheless, there were differences between the genders, inasmuch as males perceived a greater risk of relapse without chemotherapy that might be associated with their greater pessimism in facing cancer. Likewise, males exhibited less anxiety and depression than females. A classic study conducted by Pettingale KW et al. in females with early stage breast cancer found that recurrence-free survival was significantly more common among patients who had initially reacted to the cancer with denial or a fighting spirit than among patients who had responded with stoic acceptance or feelings of helplessness and hopelessness [32]. In our study, social support, optimism, and coping based on positive attitude explained less stoicism in men. Stoic individuals who perceive less social support, are less optimistic, and display a passive style of coping with illness, tend to be more withdrawn—all related with worse adjustment to their disease. In contrast, Behen JM and Rodriguez JR found no correlation between resignation, social support, and coping with illness [33]. A review of eleven studies concluded that the experience of living with cancer in old age is characterized by profound ambiguity and a predominance of social withdrawal [34]. In women, stoicism was more common in older aged individuals [1] and was associated with less perceived social support and lower levels of optimism. In line with our series, Schou I et al. found a correlation between pessimism as quantified by the LOT-R scale, maladaptive coping, and depression in their study of women with breast cancer conducted one year following surgery [35].

This study has certain limitations. First, given its cross-sectional design, a causal interpretation cannot be made of the association between stoicism and psychological variables. Longitudinal research would be necessary to test the model and examine the relationship between the variables of interest. Stoic perspective could only be assessed at a given point in time and not throughout the entire process of adapting to cancer. Secondly, we have evaluated patients' subjective appraisal of their social network without estimating the actual amount of support they received, or the support perceived by the family. Thirdly, stoicism and psychological traits were not evaluated prior to the diagnosis of cancer; as a result, the impact cancer had on them cannot be estimated.

In conclusion, stoicism was a factor that was associated with worse adaptation to cancer and was more common in males and in older participants. A stoic attitude correlated with perception of less social support, decreased optimism, and passive coping strategies (helplessness and anxious preoccupation).

## Supporting information

**S1 Table. The table shows the sex, age and patients scores on psychological scales.** (PDF)

## Acknowledgments

The authors would like to thank the investigators of the NEOcoping study, the Supportive Care Working Group of the Spanish Society of Medical Oncology (SEOM), Priscilla Chase Duran for editing the manuscript.

## Ethics approval

The study was approved by the Research Ethics Committee of the Principality of Asturias (19 January 2015) and by the Spanish Agency of Medicines and Medical Devices (AEMPS) (number to: L34LM-MM2GH-Y925U-RJDHQ).

## Author Contributions

**Conceptualization:** David Gomez, Paula Jimenez-Fonseca, Caterina Calderon.

**Data curation:** Alberto Carmona-Bayonas, Raquel Hernandez, Oliver Higuera, Jacobo Rogado, Vilma Pacheco-Barcia, María Valero, Mireia Gil-Raga, Mª Mar Muñoz, Rafael Carrión-Galindo, Caterina Calderon.

**Formal analysis:** Alberto Carmona-Bayonas, Caterina Calderon.

**Investigation:** David Gomez, Raquel Hernandez, Oliver Higuera, Jacobo Rogado, Vilma Pacheco-Barcia, María Valero, Mireia Gil-Raga, Mª Mar Muñoz, Rafael Carrión-Galindo, Paula Jimenez-Fonseca, Caterina Calderon.

**Methodology:** David Gomez, Alberto Carmona-Bayonas, Paula Jimenez-Fonseca, Caterina Calderon.

**Supervision:** Paula Jimenez-Fonseca.

**Validation:** Alberto Carmona-Bayonas.

**Writing – original draft:** David Gomez, Alberto Carmona-Bayonas, Paula Jimenez-Fonseca.

**Writing – review & editing:** David Gomez, Alberto Carmona-Bayonas, Raquel Hernandez, Oliver Higuera, Jacobo Rogado, Vilma Pacheco-Barcia, María Valero, Mireia Gil-Raga, Mª Mar Muñoz, Rafael Carrión-Galindo, Paula Jimenez-Fonseca, Caterina Calderon.

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
