## [Decision Letter · Decision Letter 0]

13 Dec 2021

PONE-D-20-38896Stoic attitude in 932 patients with cancer from the NEOcoping study: cross-sectional studyPLOS ONE

Dear Dr. Calderón,

Thank you for submitting your manuscript to PLOS ONE. After careful consideration, we feel that it has merit but does not fully meet PLOS ONE’s publication criteria as it currently stands. Therefore, we invite you to submit a revised version of the manuscript that addresses the points raised during the review process.

 Two reviewers have now evaluated your manuscript, and have identified several aspects that need careful attention in order for your submission to meet PLOS ONE's publication criteria. Please respond carefully to all of the reviewers' comments, paying particular attention to providing more detailed context to your study and clarifying and justifying the methods used.

We look forward to receiving your revised manuscript.

Kind regards,

Jamie Males

Staff Editor

PLOS ONE

Journal Requirements:

Reviewers' comments:

Reviewer's Responses to Questions

**Comments to the Author**

1. Is the manuscript technically sound, and do the data support the conclusions?

Reviewer #1: No

Reviewer #2: Partly

2. Has the statistical analysis been performed appropriately and rigorously? 

Reviewer #1: No

Reviewer #2: Yes

3. Have the authors made all data underlying the findings in their manuscript fully available?

Reviewer #1: Yes

Reviewer #2: No

4. Is the manuscript presented in an intelligible fashion and written in standard English?

Reviewer #1: Yes

Reviewer #2: No

5. Review Comments to the Author

Reviewer #1: This manuscript explored the association of stoicism with coping, psychological distress, optimism, perceived social support, and estimated risk of recurrence in cancer patients. The present study has several strengths: a well-written manuscript, clinical relevance of the study objective to the cancer survivorship and a relatively large sample size. Nevertheless, the reviewer has the following comments and criticism on the conceptual framework and methodology of the study, particularly on the incremental value of stoicism over existing cancer coping strategies.

1) Introduction: Stoicism is a interesting concept that has rarely been examined in the context of coping with cancer. The authors brought up this innovative construct and could potentially contribute to the existing literature. However, the current introduction is too brief and much more information is needed to establish the importance of the concept of stoicism in the context of cancer. The first two paragraphs are just overall introduction of stoicism without specific relevance to cancer. The authors need to provide a more focused background of the study while adding more citations from relevant literature. There are currently only 6 references in the Introduction!

2) What does the last sentence in Introduction “Finally, we have developed a model of stoicism correlating with coping, psychological stress, optimism, and perceived social support” mean? Is it one of the study objectives?

3) Given the cross-sectional nature of this study, implications of the current results are limited and measurement of the estimated risk of future recurrence might not be appropriate and relevant.

4) Mini-MAC: In page 4, the authors wrote that the Mini-MAC assesses five cognitive coping responses, but they only listed helplessness, anxious preoccupation, positive attitude, and cognitive avoidance as the 4 factors. It seems the authors have somehow adopted the 4-factor structure for Mini-MAC advocated in the Norwegian version they cited in [10]. However, as the authors have quoted, the Mini-MAC is generally perceived to have 5 factors with “positive attitude” being distinguished into fighting spirit (4 items) and 5) fatalism (5 items). The authors need to better a stronger justification not to opt for the conventional factor structure for the Mini-MAC given it is a well-established and validated scale of cancer coping.

5) Incremental value of stoicism: Following the last comment on Mini-MAC, the authors need to delineate the conceptual distinction between stoicism and existing cancer coping strategies such as fatalism and cognitive avoidance as Mini-MAC factors. It seems to the reviewer that stoicism has conceptual overlap with both fatalism and cognitive avoidance. It is essential to specify the conceptual difference between stoicism and existing cancer coping strategies to establish the need for the present study.

6) Analysis: The authors conduct tests on gender and age differences in stoicism and other psychological variables. Did they consider ANOVAs to test the potential interaction effects between age and gender? This should be feasible given the large sample size. The findings presented in Tables 2 – 4 are mostly t-tests and correlational analysis, which are rather basic and do not take full account of the complex relationships among the variables.

7) Why did the authors dichotomize the continuous stoicism score into high and low stoicism subgroups? What is the added value of conducting t-tests across the high-low subgroups over correlation as in Table 4? The results in Tables 3 and 4 appear to be redundant without new information added.

8) The authors conducted the correlation analyses in male and female subsamples separately in Table 4. Are the gender differences significant? Most of the correlations in Table 3 are rather small and < 0.30. Do they bear much clinical relevance to the participants?

9) Most of the variables in the study are coping strategies (social support, Mini-MAC, optimism). Why did the authors use these variables to predict stoicism in Table 5? Is stoicism perceived as a predictor of cancer symptoms/distress or outcome of other coping strategies in this study? Even though this study is only cross-sectional in nature and could not answer the causal direction among the study variables, it would be better if the authors could provide a clearer conceptual framework of the study variables related to the main construct of stoicism.

10) The authors wrote they conducted structural equation models for the regression analyses. However, a lot of information is missing for the SEM analysis, such as model specification, model fit, model modification, control variables in the model. How can they conduct SEM in SPSS 23.0 and have F-values in the results in SEM analysis? Presentation of the last part of results is too brief for the reviewer and readers to understand. Did the authors use latent factors of the predictors or outcomes in the SEM analysis?

Reviewer #2: In the article there is no theoretical background of concepts measured in the study, except stoic attitude. The other variables included in the research should be also described in the introduction.

In the methodological part, we have little information about specific cancer diagnosis. Cancer patients have different e. g. coping styles or depressive symptopms regarding the specific diagnosis. For example, patients may differ in depression level regarding to the diagnosis. It should be explained, why Authors of the manuscript did not expect and measure any differences in stoic attutude among patient with different diagnosis.

In the article there are some grammatical mistakes which need to be corrected, such as: p. 5 "Gender and stoicism differences on age, perception of life expectancy, and risk of toxicity, and psychological scales were measures using

independent samples t-tests." - should be ..."were measured". p. 8: "In our study, social support, optimism, and coping based on positive attitude explined less stoicism in men." - should be: "... explained...".

6. PLOS authors have the option to publish the peer review history of their article (what does this mean?). If published, this will include your full peer review and any attached files.

Reviewer #1: No

Reviewer #2: No

---

## [Author Response · Author response to Decision Letter 0]

4 Feb 2022

Reviewer #1: This manuscript explored the association of stoicism with coping, psychological distress, optimism, perceived social support, and estimated risk of recurrence in cancer patients. The present study has several strengths: a well-written manuscript, clinical relevance of the study objective to the cancer survivorship and a relatively large sample size. Nevertheless, the reviewer has the following comments and criticism on the conceptual framework and methodology of the study, particularly on the incremental value of stoicism over existing cancer coping strategies.

Well aware of the value your critical reading brings to our text, we would like to begin by thanking the reviewer for having read the text and for considering that it may be of scientific interest and for having captured the highlights and strengths of this work. Please find below our response to each of the comments made by the reviewer. 

1) Introduction: Stoicism is a interesting concept that has rarely been examined in the context of coping with cancer. The authors brought up this innovative construct and could potentially contribute to the existing literature. However, the current introduction is too brief and much more information is needed to establish the importance of the concept of stoicism in the context of cancer. The first two paragraphs are just overall introduction of stoicism without specific relevance to cancer. The authors need to provide a more focused background of the study while adding more citations from relevant literature. There are currently only 6 references in the Introduction!

In response to this question, we have expanded the explanation of the relevance and implication of stoicism in cancer patients and the available scientific evidence on this issue.

To clarify this point, the following paragraph has been added to the Introduction section:

“A study of men with prostate cancer of any stage suggests that stoicism is neither adaptive nor maladaptive for distress and quality of life outcomes and is not highly correlated with male self-esteem or psychological flexibility. The authors hypothesized that this may mean that men use stoicism to cope with prostate cancer in line with their own masculine values*. In another study by Mah et al., younger, and older patients with advanced cancer showed unique biopsychosocial correlates of pain-related stoicism and caution** Cognitive avoidance of distressing thoughts was intrapersonal, associated with greater concealment, self-doubt, and reticence in the elderly. Older adults were more likely to regulate emotions, to be less expressive about negative events and avoidance of negative information in relation to a more stoic attitude.”

*McAteer G, Gillanders D. Investigating the role of psychological flexibility, masculine self-esteem and stoicism as predictors of psychological distress and quality of life in men living with prostate cancer. Eur J Cancer Care (Engl). 2019;28(4):e13097..

**Mah K, Tran KT, Gauthier LR, et. Do Correlates of Pain-Related Stoicism and Cautiousness Differ in Younger and Older People With Advanced Cancer? J Pain. 2018;19(3):301-316. 

2) What does the last sentence in Introduction “Finally, we have developed a model of stoicism correlating with coping, psychological stress, optimism, and perceived social support” mean? Is it one of the study objectives?

The final sentence of the introduction does not correspond to either of the two objectives mentioned above and has therefore been deleted.

3) Given the cross-sectional nature of this study, implications of the current results are limited and measurement of the estimated risk of future recurrence might not be appropriate and relevant.

As the reviewer refers, cross-sectional nature of this study is a limiting factor and this is stated in the first limitation in the discussion section: "First, given its cross-sectional design, a causal interpretation cannot be made of the association between stoicism and psychological variables. Stoic perspective could only be assessed at a given point in time and not throughout the entire process of adapting to cancer.”

Patients were asked about their perceived risk of recurrence before starting chemotherapy adjuvant/complementary to surgery with curative intent. Since tumor stage was similar, the influence of stoicism, gender and other psychological variables on patients' perception of their risk of cancer recurrence was assessed. 

Therefore, the patient's perception of their risk of cancer recurrence was considered, but no calculation of the risk of recurrence was made based on any of the variables studied. Therefore, the analyses do not establish causal relationships in oncological outcomes.

4) Mini-MAC: In page 4, the authors wrote that the Mini-MAC assesses five cognitive coping responses, but they only listed helplessness, anxious preoccupation, positive attitude, and cognitive avoidance as the 4 factors. It seems the authors have somehow adopted the 4-factor structure for Mini-MAC advocated in the Norwegian version they cited in [10]. However, as the authors have quoted, the Mini-MAC is generally perceived to have 5 factors with “positive attitude” being distinguished into fighting spirit (4 items) and 5) fatalism (5 items). The authors need to better a stronger justification not to opt for the conventional factor structure for the Mini-MAC given it is a well-established and validated scale of cancer coping.

The authors of this paper validated the Spanish version of the Mini-MAC in a Spanish series of cancer patients*.

The results of the factor analysis indicated a 4-factor structure. Three subscales have similar psychometric properties to the Helplessness, Anxious preoccupation, and Cognitive avoidance subscales of the original Mini-MAC and the Fighting Spirit and Fatalism subscales were combined into the Positive Attitude scale. The scale scores derived from the four factors showed acceptable accuracy for individual measures, as well as stability across test-retest assessments at 6 months. Validity assessments found meaningful relations between the derived scale scores, and Brief Symptom Inventory (BSI) depression and anxiety scores and Functional Assessment of Chronic Illness Therapy (FACIT) spiritual well-being scores.

Therefore, given that our team had validated and confirmed in previous work the accuracy of a 4-factor based scale, it was decided to use it in this work.

In the method section, we have added the following clarification regarding this point:

“The 29-item mini-MAC assesses five cognitive coping responses: helplessness, anxious preoccupation, fighting spirit, fatalism and cognitive avoidance. For this study, the version with a 4-factor structure validated in Spanish was used, grouping fighting spirit and fatalism into a single factor, positive attitude*.”

*Calderon C, Lorenzo-Seva U, Ferrando PJ, et al. Psychometric properties of Spanish version of the Mini-Mental Adjustment to Cancer Scale. Int J Clin Health Psychol. 2021;21(1):100185. doi: 10.1016/j.ijchp.2020.06.001.

5) Incremental value of stoicism: Following the last comment on Mini-MAC, the authors need to delineate the conceptual distinction between stoicism and existing cancer coping strategies such as fatalism and cognitive avoidance as Mini-MAC factors. It seems to the reviewer that stoicism has conceptual overlap with both fatalism and cognitive avoidance. It is essential to specify the conceptual difference between stoicism and existing cancer coping strategies to establish the need for the present study.

The term fatalism refers to the belief in the determinism of events, directed by causes independent of human will. This notion of fatalism carries a negative connotation. While Stoicism is based on the notion of control over life-disturbing emotions, the goal is based on tolerance and self-control. We have made changes to the introduction to delineate the concepts.

6) Analysis: The authors conduct tests on gender and age differences in stoicism and other psychological variables. Did they consider ANOVAs to test the potential interaction effects between age and gender? This should be feasible given the large sample size. The findings presented in Tables 2 – 4 are mostly t-tests and correlational analysis, which are rather basic and do not take full account of the complex relationships among the variables.

You are right, it would be interesting to analyze the interaction between stoicism and sex and age and we will take that into account for future studies. The objective of the present study was to analyze the differences between patients with high and low levels of stoicism.

7) Why did the authors dichotomize the continuous stoicism score into high and low stoicism subgroups? What is the added value of conducting t-tests across the high-low subgroups over correlation as in Table 4? The results in Tables 3 and 4 appear to be redundant without new information added.

To carry out the statistical analyzes presented in tables 2 and 4, we rely on the scientific literature that indicates that significant differences are observed between men and women in stoicism. The correlation of stoicism with the psychological variables according to sex was carried out first, in order to know which of these would be predictive in explaining stoicism (Table 5), also differentiating it by sex.

8) The authors conducted the correlation analyses in male and female subsamples separately in Table 4. Are the gender differences significant? Most of the correlations in Table 3 are rather small and < 0.30. Do they bear much clinical relevance to the participants?

To carry out the statistical analyzes presented in tables 2 and 4, we rely on the scientific literature that indicates that significant differences are observed between men and women in stoicism. The correlation of stoicism with the psychological variables according to sex was carried out first, in order to know which of these would be predictive in explaining stoicism (Table 5), also differentiating it by sex.

9) Most of the variables in the study are coping strategies (social support, Mini-MAC, optimism). Why did the authors use these variables to predict stoicism in Table 5? Is stoicism perceived as a predictor of cancer symptoms/distress or outcome of other coping strategies in this study? Even though this study is only cross-sectional in nature and could not answer the causal direction among the study variables, it would be better if the authors could provide a clearer conceptual framework of the study variables related to the main construct of stoicism.

Our study aims to explain the percentage of explained variance of various psychological variables in stoicism, it does not try to explain how stoicism develops, for this we would need to carry out a longitudinal study. Following your recommendations, we have improved the conceptual framework of the introduction.

10) The authors wrote they conducted structural equation models for the regression analyses. However, a lot of information is missing for the SEM analysis, such as model specification, model fit, model modification, control variables in the model. How can they conduct SEM in SPSS 23.0 and have F-values in the results in SEM analysis? Presentation of the last part of results is too brief for the reviewer and readers to understand. Did the authors use latent factors of the predictors or outcomes in the SEM analysis?

As described in Data Analysis, in the manuscript we use two linear regression analyzes to predict the association between stoicism and psychological scales in men and women. In subsequent statistical analyses, stoicism served as the criterion variable in regression analyses, while social support (according to the MSPSS), coping strategies (Mini-MAC), psychological distress (BSI-18) and optimism (LOT-R) were used as predictors.

We have not used SEM. We are sorry.

 

Reviewer #2: In the article there is no theoretical background of concepts measured in the study, except stoic attitude. The other variables included in the research should be also described in the introduction.

In response to the reviewer's request, we have added the following paragraph in the introduction section relating to the other items, coping, psychological distress, optimism, and perception of social support:

“As cancer is a stressful event associated with fear and prognostic uncertainty, it is common for patients to employ coping strategies to deal with it. Stoicism and emotional distress (anxiety, depression, and somatization) may interfere with coping by increasing anxious preoccupation*. Dispositional optimism may act as a resource that maintains a positive mood, provides a more flexible coping style to face with the inability to control stressful stimuli, and protects patients from the possible negative effects of cancer treatment and cancer itself in case of recurrence**,***. Likewise, social support, defined as the perception that others are willing to help, appears to operate similarly to optimism and is a critical factor in decreasing negative psychological affect in individuals with cancer****.”

*Jimenez-Fonseca P, Calderón C, Hernández R, et al. Factors associated with anxiety and depression in cancer patients prior to initiating adjuvant therapy. Clin Transl Oncol. 2018;20(11):1408-1415. doi: 10.1007/s12094-018-1873-9.

**Ciria-Suarez L, Calderon C, Fernández Montes A, et al. Optimism and social support as contributing factors to spirituality in Cancer patients. Support Care Cancer. 2021;29(6):3367-3373. doi: 10.1007/s00520-020-05954-4.

***Ciria-Suarez L, Jiménez-Fonseca P, Palacín-Lois M, et al. Breast cancer patient experiences through a journey map: A qualitative study. PLoS One. 2021 Sep 22;16(9):e0257680. doi: 10.1371/journal.pone.0257680.

****Calderon C, Gomez D, Carmona-Bayonas A, et al. Social support, coping strategies and sociodemographic factors in women with breast cancer. Clin Transl Oncol. 2021;23(9):1955-1960. doi: 10.1007/s12094-021-02592-y

In the methodological part, we have little information about specific cancer diagnosis. Cancer patients have different e. g. coping styles or depressive symptoms regarding the specific diagnosis. For example, patients may differ in depression level regarding to the diagnosis. It should be explained, why Authors of the manuscript did not expect and measure any differences in stoic attitude among patient with different diagnosis.

Analyzing the differences in depression and coping strategies among cancer patients is very interesting, but it is not the objective of our study.

In the article there are some grammatical mistakes which need to be corrected, such as: p. 5 "Gender and stoicism differences on age, perception of life expectancy, and risk of toxicity, and psychological scales were measures using

independent samples t-tests." - should be ..."were measured". p. 8: "In our study, social support, optimism, and coping based on positive attitude explined less stoicism in men." - should be: "... explained...".

Following the reviewer's indications, we have checked the text in detail and corrected the two spelling mistakes and one other error that we have detected.

We have highlighted them in blue in the text.

Cognizant of the contribution their insightful and critical observations have made to enhancing the quality of the text, the researchers of this study would again like to thank the Editor and reviewers for their time and effort.

---

## [Decision Letter · Decision Letter 1]

18 Mar 2022

PONE-D-20-38896R1Stoic attitude in patients with cancer from the NEOcoping study: cross-sectional studyPLOS ONE

Dear Dr. Calderón,

Thank you for submitting your manuscript to PLOS ONE. After careful consideration, we feel that it has merit but does not fully meet PLOS ONE’s publication criteria as it currently stands. Therefore, we invite you to submit a revised version of the manuscript that addresses the points raised during the review process.

To preserve transparency and uphold the integrity of the scientific process, I acknowledge my previous reviewer role in this decision letter to the authors. We were fortunate to have the insightful comments from four expert reviewers on the revised manuscript. The authors have in general conducted a satisfactory revision on the manuscript in view of the previous reviewer comments. However, the manuscript still needs to address the methodological concerns raised by Reviewer 3 regarding statistical analysis. As Reviewer 3 pointed out, it is necessary to carry out and include some additional statistical methods of greater capacity to predict more complex relational maps.

We look forward to receiving your revised manuscript.

Kind regards,

Ted C.T. Fong

Academic Editor

PLOS ONE

Reviewers' comments:

Reviewer's Responses to Questions

**Comments to the Author**

1. If the authors have adequately addressed your comments raised in a previous round of review and you feel that this manuscript is now acceptable for publication, you may indicate that here to bypass the “Comments to the Author” section, enter your conflict of interest statement in the “Confidential to Editor” section, and submit your "Accept" recommendation.

Reviewer #2: (No Response)

Reviewer #3: (No Response)

Reviewer #4: (No Response)

Reviewer #5: All comments have been addressed

2. Is the manuscript technically sound, and do the data support the conclusions?

Reviewer #2: Yes

Reviewer #3: Partly

Reviewer #4: Partly

Reviewer #5: Yes

3. Has the statistical analysis been performed appropriately and rigorously? 

Reviewer #2: Yes

Reviewer #3: No

Reviewer #4: Yes

Reviewer #5: Yes

4. Have the authors made all data underlying the findings in their manuscript fully available?

Reviewer #2: Yes

Reviewer #3: Yes

Reviewer #4: Yes

Reviewer #5: Yes

5. Is the manuscript presented in an intelligible fashion and written in standard English?

Reviewer #2: No

Reviewer #3: Yes

Reviewer #4: Yes

Reviewer #5: Yes

6. Review Comments to the Author

Reviewer #2: The article has been improved but there are still some issues to explain.

The Authors still did not refer to the question: "...why Authors of the manuscript did not expect and measure any differences

in stoic attitude among patients with different diagnosis? (diagnosis of type of cancer of course).

There are still many linguistic errors and the style should be improved (e.g. in the sentense: "Stoicism has been associated with being male, and Wagstaff and Rowledge found that men exhibited higher stoicism scores; nevertheless, no studies on the associations with stoicism specifically in men and women with cancer").

Reviewer #3: The manuscript addresses a topic that is consistent with the scope and aims of this Journal

It contributes new knowledge to the specialty in an important issue for health.

Writing style, organization, and clarity is adequate.

The study shows new findings.

The use of language is clear and precise.

The ideas are presented in an economical way.

The content is interesting.

The content is well organized with logical flow.

The work is grounded in recently published literature.

The purpose of the manuscript is important for health:

the aim was to explore associations of stoicism with other outcomes in patients with non-metastatic cancer, and assess gender differences in age, perception of the risk of cancer relapse, the risk of adjuvant chemotherapy-related toxicity, coping, psychological stress, optimism, and perceived social support.

The sample appears to be adequate with appropriate data:

In general, the scales in this study have good internal consistency.

Internal consistency was analyzed using Cronbach’s alpha coefficient. Values of α above 0.70 were considered adequate, confirming that the items were sufficiently correlated

The statistical analyses performed are adequate but too simple (frequency, correlation, comparision of means, regression) to generate a novel impact or significant conclusions from the research question posed. It would be necessary to consider a greater impact of the data from analyses that allow more predictive conclusions.

They informed that the margin error of the sample size was calculated (Cohen’s statistis).. The effect size has to be indicated as a relevant information in comparison of means in order to be able to properly interpret the statistical power of the result found.

Statistical analyses based on regression are adequate for prediction but trying a Path analysis, SEM or QCA analysis could improve the prediction capacity with such a big sample and could answer the predicting objectives of this study in a more complete way.

A structural equation model (SEM) is needed for these prediction objectives and this big size of sample. Particularly, the structural equation model, (SEM) the estimate provided by the robust method of maximum likelihood estimation (ML), recommended to correct the possible absence of multivariate normality, is recommended to be applied in all cases. A structural equation model could be done to predict dependent variables through independent variables. In addition, multigroup have to be conducted to test the moderating effect of variables such as gender, type of cancer, severity of the disease (measured by variables such as prognosis, cancer stage, risk of cancer recurrence, toxicity)

For example, in consideration of the dropout rate according to the criteria of Bae (2017) at least 200 participants required in path analysis with 12 or fewer observation. In this study we have a sample over 900 participants.

The QCA allows the quantitative analysis even in cases of a small number of participants (that is not particularly the case of this study), but using Boolean algebra as a formal tool to identify which of a series of factors (independent variables or causal conditions) are associated with the presence of a given result (criterion variable or result condition). Thus, it allows proposing pathways (which combine a particular interaction between the variables) to optimize the prediction of the independent variable.

It is necessary to carry out and include some additional statistical method of greater capacity to predict more complex relational maps.

Reviewer #4: This is an interesting study that will start an important conversation about the role of stocism in coping with cancer. There are several weaknesses that need to be addressed:

1. The introductory section does not really create the right background to the paper even though it was amended from the original. The authors need to present an argument that provides the basis for exploring stocism in cancer.

2. This is a substudy of a larger study I take it. That said the aims of the larger study are not articulated and there are no aims for this study or research questions that guided the analysis.

3. The lack of an aim and research questions limits that discussion of findings and is a weakness. I was also disappointed that the discussion did not point to the potential clinical relevance of the construct, how you might assess in practice or point to the sort of research that is now needed if the construct has merit for further investigation.

I have also made some notes on the manuscript as at times the authors make comments that suggest a causal or directional relationship that the study is not set up to demonstrate.

Reviewer #5: This manuscript entitled Stoic attitude in patients with cancer from the NEOcoping study: cross-sectional study aimed to explore the association of stoicism with coping, psychological distress, optimism, perceived social support, and estimated risk of recurrence in cancer patients. I congratulation the authors for this well-written manuscript that covers an importante subject that may guide care for cancer survivors. The revised version is much better than the original one. I have only minor concerns related to some points that I listed below

The aim of this study should be the same in the abstract and at the end of the introduction.

It was stated in method that 932 individuals were recruited. It should be mentioned in methods how many participants were included and whow many were male or female.

7. PLOS authors have the option to publish the peer review history of their article (what does this mean?). If published, this will include your full peer review and any attached files.

Reviewer #2: No

Reviewer #3: No

Reviewer #4: No

Reviewer #5: No

---

## [Author Response · Author response to Decision Letter 1]

3 May 2022

Reviewers' comments:

Reviewer #2: The article has been improved but there are still some issues to explain.

The Authors still did not refer to the question: "...why Authors of the manuscript did not expect and measure any differences in stoic attitude among patients with different diagnosis? (diagnosis of type of cancer of course).

Comment: Following your suggestion, we have performed the calculation and the results indicate that there are differences depending on the primary tumor (F= 10.445, p=0.001). Patients with colon cancer score higher on stoicism than patients with breast cancer (M: 57.4 vs M 53.8, IJ: 3.544, p=0.001, respectively). 

-Colon (M: 57.4, SD: 7.3)

- Breast (M: 53.8, SD: 7.9)

- Stomach (M: 56.8; SD: 7.1)

- Others (M: 55.9; SD: 7.8)

We have included these in the manuscript (result section): 

“In terms of primary cancer, patients with colon cancer scored higher on stoicism than patients with breast cancer (M: 57.4 vs M 53.8; F= 10.445, p=0.001; IJ: 3.544, p=0.001, respectively).”

There are still many linguistic errors and the style should be improved (e.g. in the sentense: "Stoicism has been associated with being male, and Wagstaff and Rowledge found that men exhibited higher stoicism scores; nevertheless, no studies on the associations with stoicism specifically in men and women with cancer").

Comment: The text was translated by a professional translator who has revised it again after the response to the queries, making modifications. In particular, this sentence has been removed from the text. The authors hope that this new version of the article will be better written for the reviewer.

Reviewer #3: The manuscript addresses a topic that is consistent with the scope and aims of this Journal. The purpose of the manuscript is important for health: the aim was to explore associations of stoicism with other outcomes in patients with non-metastatic cancer, and assess gender differences in age, perception of the risk of cancer relapse, the risk of adjuvant chemotherapy-related toxicity, coping, psychological stress, optimism, and perceived social support.

The statistical analyses performed are adequate but too simple (frequency, correlation, comparison of means, regression) to generate a novel impact or significant conclusions from the research question posed. It would be necessary to consider a greater impact of the data from analyses that allow more predictive conclusions. They informed that the margin error of the sample size was calculated (Cohen’s statistics). The effect size has to be indicated as a relevant information in comparison of means in order to be able to properly interpret the statistical power of the result found.

Comment: 

We thank the reviewer for this suggestion, which has helped to increase the rigor and precision of the results.

Following this suggestion, we have included information on how to interpret the statistical power of the analyses performed and on the use of a Structural Equation Modeling (SEM) to estimate the correlation between variables. This is the text added in the methods section:

“Effect sizes were calculated by Cohen’s statistics. Cohen’s d was reported as an indicator of the effect size of the differences, with d > 0.2 representing a small, d > 0.5 a medium and d > 0.8 a large effect size [27]. Bivariate correlations were calculated between stoicism (LSS) and psychological scales (MSPSS, Mini-Mac, BSI-18, and LOT-R) and sociodemographic factors. All data were checked for normality, outliers, and the assumptions of multicollinearity and homoscedasticity. Structural Equation Modeling (SEM) is a method for building, estimating, and testing theoretical models of the relationships between variables. It can be used instead of multiple regression and other methods to analyze the strength of correlations between individual variable indicators in a specific population [28]. In this study, the previously determined significant factors were used in the SEM to identify the relationship between stoicism and psychological scales. Standardized direct, indirect, and total effects with corresponding 95% bias-corrected confidence intervals (CI) were measure using the bootstrapping methods [28,29]. The model fit was tested using the normed χ2 value (NC; desired value < 2.0, desired significance P > 0.05), the goodness-of-fit index; the Comparative Fit Index (CFI), the Tucker-Lewis index (TLI), the Normed Fit index (NFI) (>0.95 indicating an excellent fit); root mean square of approximation (RMSEA; desired value< 0.06) [28].”

Statistical analyses based on regression are adequate for prediction but trying a Path analysis, SEM or QCA analysis could improve the prediction capacity with such a big sample and could answer the predicting objectives of this study in a more complete way. A structural equation model (SEM) is needed for these prediction objectives and this big size of sample. Particularly, the structural equation model, (SEM) the estimate provided by the robust method of maximum likelihood estimation (ML), recommended to correct the possible absence of multivariate normality, is recommended to be applied in all cases. A structural equation model could be done to predict dependent variables through independent variables. In addition, multigroup have to be conducted to test the moderating effect of variables such as gender, type of cancer, severity of the disease (measured by variables such as prognosis, cancer stage, risk of cancer recurrence, toxicity). For example, in consideration of the dropout rate according to the criteria of Bae (2017) at least 200 participants required in path analysis with 12 or fewer observation. In this study we have a sample over 900 participants.

The QCA allows the quantitative analysis even in cases of a small number of participants (that is not particularly the case of this study) but using Boolean algebra as a formal tool to identify which of a series of factors (independent variables or causal conditions) are associated with the presence of a given result (criterion variable or result condition). Thus, it allows proposing pathways (which combine a particular interaction between the variables) to optimize the prediction of the independent variable. It is necessary to carry out and include some additional statistical method of greater capacity to predict more complex relational maps.

Comment: As suggested, we have adapted the stoicism theoretical model of Furnham et al., 2003; Spiers, 2006, described in the introduction. This stoicism model represents the relationship between several key constructs among stoicism, social support, optimism, and coping strategies. 

We presume that: a) social support and optimism will positively affect patients' stoicism and coping strategies; b) stoicism will be related to coping strategies; c) and the relationships between social support and optimism and coping strategies will be mediated by stoicism. 

We have also carried out additional statistical analyses (using SEM) to predict this model and have modified the last results section by adding a new text and two new figures:

“Trajectory analysis results: relationship among stoicism and psychological variables 

In men, as displayed in Figure 1, social support and optimism were directly, and negatively associated with stoicism (β =-0.17 and β =-0.23, respectively). Stoicism was directly, and positively associated with anxious preoccupation (β =0.12). Social support was positively associated with positive aptitude (β =0.17), and depression was positively associated with anxious preoccupation (β =0.56). The model fitted the data perfectly, X2=11.318, p=0.045; CFI=0.97; TLI=0.95; NFI=0.96; RMSEA=0.05 (90% CI [0.001, 0.074]). 

In women, as displayed in Figure 2, stoicism was directly, and negatively associated with social support and positively with age (β =-0.21 and β =0.21, respectively). Optimism was directly, and negatively associated with stoicism (β =-0.14), and positively with anxious preoccupation and helplessness (β =0.26 and β =-0.30, respectively). Stoicism was directly, and positively associated with helplessness (β =0.12). The model fitted the data perfectly, X2= 14.447, p=0.044; CFI= 0.98; TLI= 0.95; NFI= 0.97M RMSEA= 0.04 (90% CI [0.000, 0.064]).”

Figure 1. Tested study model: relationship between stoicism and psychological variables in men. Note: Dashed lines indicate statistically non-significant paths. *p<0.01; **p<0.001.

 Figure 2. Tested study model: relationship between stoicism and psychological variables in women. 

Note: Dashed lines indicate statistically non-significant paths. *p<0.01; **p<0.001.

Reviewer #4: This is an interesting study that will start an important conversation about the role of stocism in coping with cancer. There are several weaknesses that need to be addressed:

1. The introductory section does not really create the right background to the paper even though it was amended from the original. The authors need to present an argument that provides the basis for exploring stocism in cancer.

Comment: we have modified several sections of the introduction to present a robust and well-argued theoretical framework to justify the study of the influence of stoicism on psychological variables associated with coping with cancer. 

At the end of paragraph 2 of the introduction we have added this text:

“Some studies have shown that stoicism would be very sensitive to the health setting, and this could have implications for the patient's treatment and prognosis [6, 7]. Maintaining a stoic attitude towards pain and knowing one's own limits could be particularly beneficial for the patient and allow them to adopt a more adaptive coping style [4], or conversely, the stoic attitude could restrict attempts at medical intervention if the patient does not report their real clinical situation (i. e., the patient feels pain, but does not show it). Therefore, according to other researchers, stoic characteristics hinder seeking care, delay diagnosis, and prevent the implementation of improvements in psychological care [1].”

At the end of the introduction, we have added a fourth paragraph: 

“Overall, the interrelationships between social support, optimism, stoicism, coping strategies and psychological distress remain to be further investigated. In the oncology field, little has been explored about the pathways through which stoicism influences coping in cancer patients. Some authors suggest that men and older people have higher levels of stoicism than women [2,17] or younger people [1]. Other authors have suggested that men and older adults are more stoic because they find it more difficult to identify and express their emotions [18], which would have a negative influence on seeking psychological help [19]. Meanwhile, other research suggests that the stoic attitude is more related to the ability to self-regulate one's emotions [12].”

2. This is a substudy of a larger study I take it. That said the aims of the larger study are not articulated and there are no aims for this study or research questions that guided the analysis.

Comment: In response to this comment, we have included the following text with the hypotheses to make it easier to read and to better express what we expect to find:

“In this perspective, our study has assessed the fit of an adapted stoicism model [4,5] (figure 1) using data collected from a large sample of patients who had cancer resected with curative intent. This adapted model of stoicism represents key constructs (social support, optimism, psychological distress, stoicism, and coping strategies), as well as the relationships between the variables used to operationalize the constructs. We presume that: a) social support and optimism will positively affect stoicism and coping strategies; b) stoicism will be related to coping strategies; and c) the relationships between social support, optimism and coping strategies will be mediated by stoicism.”

3. The lack of an aim and research questions limits that discussion of findings and is a weakness. I was also disappointed that the discussion did not point to the potential clinical relevance of the construct, how you might assess in practice or point to the sort of research that is now needed if the construct has merit for further investigation.

Comment: We hope that the manuscript has been improved by the rewriting of the theoretical framework in the introduction, the inclusion of concise objectives and hypotheses, and explanations in the discussion that make clear the relevance, pertinence and applicability of this research. We have also included the prevalence of stoicism:

“Overall, 27% of patients had low (PD≤ 46), and 33% had high stoicism scores (PD≥64).”

I have also made some notes on the manuscript as at times the authors make comments that suggest a causal or directional relationship that the study is not set up to demonstrate.

Comment: we have included a new analysis with Structural Equation Modeling (SEM) to support the assertions we made in the manuscript and which we have explained in detail in the questions raised by reviewer #3. This is the text added to the statistical method section:

“In this study, the previously determined significant factors were used in the SEM to identify the relationship between stoicism and psychological scales. Standardized direct, indirect, and total effects with corresponding 95% bias-corrected confidence intervals (CI) were measure using the bootstrapping methods [28,29]. The model fit was tested using the normed χ2 value (NC; desired value < 2.0, desired significance P > 0.05), the goodness-of-fit index; the Comparative Fit Index (CFI), the Tucker-Lewis index (TLI), the Normed Fit index (NFI) (>0.95 indicating an excellent fit); root mean square of approximation (RMSEA; desired value< 0.06) [28].”

Reviewer #5: The revised version is much better than the original one. I have only minor concerns related to some points that I listed below. The aim of this study should be the same in the abstract and at the end of the introduction.

It was stated in method that 932 individuals were recruited. It should be mentioned in methods how many participants were included and who many were male or female.

Comment: we have reviewed the objective of the abstract and the text and adjusted them to match. We have added how many patients were women and men through the following text (results section): “Mean age was 62.1 years (standard deviation [SD]=12.1, range 24-85), 61% (n=566) were female and 39% (n=366) were male.”

Cognizant of the contribution their insightful and critical observations have made to enhancing the quality of the text, the researchers of this study would again like to thank the Editor and reviewers for their time and effort.

---

## [Decision Letter · Decision Letter 2]

27 May 2022

Stoic attitude in patients with cancer from the NEOcoping study: cross-sectional study

PONE-D-20-38896R2

Dear Dr. Calderón,

We’re pleased to inform you that your manuscript has been judged scientifically suitable for publication and will be formally accepted for publication once it meets all outstanding technical requirements.

Kind regards,

Ted C.T. Fong

Guest Editor

PLOS ONE

Additional Editor Comments (optional):

Please address the remaining comment from Reviewer 4 and revise the abstract to reflect the altered analysis. 

Reviewers' comments:

Reviewer's Responses to Questions

**Comments to the Author**

1. If the authors have adequately addressed your comments raised in a previous round of review and you feel that this manuscript is now acceptable for publication, you may indicate that here to bypass the “Comments to the Author” section, enter your conflict of interest statement in the “Confidential to Editor” section, and submit your "Accept" recommendation.

Reviewer #2: All comments have been addressed

Reviewer #3: All comments have been addressed

Reviewer #4: All comments have been addressed

Reviewer #5: All comments have been addressed

2. Is the manuscript technically sound, and do the data support the conclusions?

Reviewer #2: (No Response)

Reviewer #3: Yes

Reviewer #4: Yes

Reviewer #5: Yes

3. Has the statistical analysis been performed appropriately and rigorously? 

Reviewer #2: (No Response)

Reviewer #3: Yes

Reviewer #4: Yes

Reviewer #5: Yes

4. Have the authors made all data underlying the findings in their manuscript fully available?

Reviewer #2: (No Response)

Reviewer #3: Yes

Reviewer #4: Yes

Reviewer #5: Yes

5. Is the manuscript presented in an intelligible fashion and written in standard English?

Reviewer #2: (No Response)

Reviewer #3: Yes

Reviewer #4: Yes

Reviewer #5: Yes

6. Review Comments to the Author

Reviewer #2: (No Response)

Reviewer #3: All comments have been addressed, I recommend "Accept".

The effect size has been indicated as a relevant information in comparison of means in order to be able to properly interpret the statistical power of the result found. They have included information on how to interpret all the statistical power of the analyses performed. They have done new anaylisis, a Structural Equation Modeling (SEM) to estimate the correlation between variables.

Reviewer #4: The abstract needs to be revised to address the altered analysis.

Otherwise the authors have addressed reviewer comments.

Reviewer #5: This manuscript entitled Stoic attitude in patients with cancer from the NEOcoping study: cross-sectional study aimed to explore the association of stoicism with coping, psychological distress, optimism, perceived social support, and estimated risk of recurrence in cancer patients. The authors responded to all queries. I congratulate them for this great work

7. PLOS authors have the option to publish the peer review history of their article (what does this mean?). If published, this will include your full peer review and any attached files.

Reviewer #2: No

Reviewer #3: No

Reviewer #4: No

Reviewer #5: No

---

## [Editor Report · Acceptance letter]

8 Jul 2022

PONE-D-20-38896R2 

Stoic attitude in patients with cancer from the NEOcoping study: cross-sectional study 

Dear Dr. Calderon:

I'm pleased to inform you that your manuscript has been deemed suitable for publication in PLOS ONE. Congratulations! Your manuscript is now with our production department. 

Kind regards, 

on behalf of

Dr. Ted C.T. Fong 

Guest Editor

PLOS ONE